# Revealing Interconnections between Diseases: from Statistical Methods to Large Language Models

## Abstract

Identifying disease interconnections through manual analysis of large-scale clinical data is labor-intensive, subjective, and prone to expert disagreement. While machine learning (ML) shows promise, three critical challenges remain: (1) selecting optimal methods from the vast ML landscape, (2) determining whether real-world clinical data (e.g., electronic health records, EHRs) or structured disease descriptions yield more reliable insights, (3) the lack of "ground truth," as some disease interconnections remain unexplored in medicine. Large language models (LLMs) demonstrate broad utility, yet they often lack specialized medical knowledge. To address these gaps, we conduct a systematic evaluation of seven approaches for uncovering disease relationships based on two data sources: (i) sequences of ICD-10[1] codes from MIMIC-IV EHRs and (ii) the full set of ICD-10 codes, both with and without textual descriptions. Our framework integrates the following: (i) a statistical co-occurrence analysis and a masked language modeling (MLM) approach using real clinical data; (ii) domain-specific BERT variants (Med-BERT and BioClinicalBERT); (iii) a general-purpose BERT and document retrieval; and (iv) four LLMs (Mistral, DeepSeek, Qwen, and YandexGPT). Our graph-based comparison of the obtained interconnection matrices shows that the LLM-based approach produces interconnections with the lowest diversity of ICD code connections to different diseases compared to other methods, including text-based and domain-based approaches. This suggests an important implication: LLMs have limited potential for discovering new interconnections. In the absence of ground truth databases for medical interconnections between ICD codes, our results constitute a valuable medical disease ontology that can serve as a foundational resource for future clinical research and artificial intelligence applications in healthcare. The results are available in our GitHub repository[2]

## 1 Introduction

Electronic health records (EHRs) provide a valuable resource for studying disease progression and relationships between diagnoses. However, analyzing such data manually is not feasible due to its large volume and complexity, especially when dealing with conditions like cancer that often progress without clear symptoms [3][4]. Moreover, experts' examination is a subjective process, characterized by inter-rater variability.

Machine learning (ML) can help discover hidden patterns in medical data, but many existing models are hard to interpret. In particular, it is not always clear whether large language models (LLMs) make predictions based on meaningful medical knowledge or simply rely on textual similarities between diagnosis descriptions (Cui et al., 2025). This is especially critical in healthcare, where model decisions must align with established medical knowledge and pathophysiological mechanisms.

---

[1] International Classification of Diseases, 10th Revision (ICD-10), https://icd.who.int/browse10/2019/en

[2] https://anonymous.4open.science/r/medical-disease-ontology/

[3] https://combatcancer.com/the-challenges-of-asymptomatic-silent-cancer

[4] https://www.vinmec.com/eng/blog/how-long-does-it-take-for-cancer-to-progress-without-you-even-knowing-it-en

To address this gap, we compare different ways of obtaining diseases interconnections' scores. We also analyze and compare the obtained results and summarize it into medical disease ontology. **Our contribution is the following:**

1. We provide an interconnections between diseases using 10 different approaches: (real database-based) Fisher's exact test, Jaccard similarity, and a masked language modeling (MLM); (models pretrained on medical domain) pretrained Med-BERT and BioClinicalBERT; (text-based approaches) pretrained BERT and Yandex Doc Search; (LLMs) DeepSeek, Qwen, and YandexGPT.

2. We conduct an analysis of the resulting interconnection matrices, including visual analysis of interconnection matrices, t-SNE visualization, graph-based comparison including graphs' degrees analysis, PR AUC calculation and assessment without real ground truth.

3. We aggregate all inferred disease interconnections based on the number of methods that independently identify them, thereby constructing a consensus medical ontology. Interconnections consistently recovered by a large number of diverse models are assigned higher confidence and are more likely to reflect established clinical relationships. Conversely, those supported by only a few methods may represent novel or previously underreported associations, offering promising hypotheses for further clinical investigation.

4. We analyze manually some of the revealed cases using the existing medical literature. These confirm the correctness of the utilized methods and their ability for further diseases interconnections' examination.

All the codes and materials, including interconnection matrices, figures and the ontology are provided in our GitHub[5].

## 2 RELATED WORK

Predictive modeling with EHRs has advanced from early temporal models to transformer-based and pretrained representations, improving how disease interconnections are identified. Key challenges include heterogeneous data formats (structured codes, free text, numerical measurements) and scarce labels for rare outcomes, motivating the use of robust pretrained models (Shickel et al., 2018; Wang et al., 2024).

Models extracting social determinants of health (SDoH), such as Flan-T5, accurately identify employment, housing, and social support, outperforming rule-based systems (Guevara et al., 2024). LLMs have also been applied to de-identify clinical text and normalize temporal data, enabling secondary EHR use while preserving privacy (Dai et al., 2025).

Large-scale generative models extend these advances. Delphi-2M, trained on UK Biobank and validated on Danish cohorts, predicts over 1000 diseases and simulates long-term health trajectories (Shmatko et al., 2025). Similarly, *Foresight* models entire timelines to synthesize plausible future events (Kraljevic et al., 2024).

Transformer architectures for sequential EHR modeling (e.g., BEHRT, Med-BERT) improve downstream prediction and transfer to smaller cohorts by encoding event order and patient-specific context (Li et al., 2020; Rasmy et al., 2020). More recent extensions such as ExBEHRT and CEHR-BERT explore disease subtypes and temporal structure (Rupp et al., 2023; Pang et al., 2021). Graph-augmented transformers further enrich representations with structural priors like code hierarchies or knowledge graphs, enhancing tasks such as medication recommendation and disease progression modeling (Shang et al., 2019).

Statistical baselines remain relevant. For example, (Fotouhi et al., 2018) compared network construction methods (OER, disparity filter, link salience) to uncover distinct comorbidity patterns.

In parallel, statistical methods for constructing disease comorbidity networks have remained important baselines. For example, (Fotouhi et al., 2018) compared network construction methods (OER, disparity filter, link salience) to uncover distinct comorbidity patterns.

---

[5]https://anonymous.4open.science/r/medical-disease-ontology/

Emerging directions explore hybrid integration of episodic memory and knowledge graphs with transformers (e.g., AriGraph), enabling structured reasoning beyond co-occurrence and temporal analysis (Anokhin et al., 2024).

Overall, the field has progressed from statistical networks and EHR-specific transformers to graph-augmented and generative models. Systematic comparative studies evaluating statistical baselines, EHR transformers, LLM embeddings, and generative approaches under a unified framework remain scarce. Moreover, it remains unclear whether the relationships identified by text-based LLMs reflect real clinical patterns or are artifacts of textual similarity. Our work fills this gap by conducting a comprehensive comparison of six different categories of methods on a single dataset.

## 3 METHODOLOGY

### 3.1 DATA OVERVIEW

#### 3.1.1 REAL-WORLD DATA

**Data description** The Medical Information Mart for Intensive Care (MIMIC) is a family of publicly available, real-world clinical datasets developed by the Massachusetts Institute of Technology and hosted on the PhysioNet platform [6]. These resources contain rich, structured information from electronic health records of patients admitted to intensive care units, making them invaluable for clinical research and the development of medical AI applications.

For this study, we use real-world clinical data from the MIMIC-IV dataset, the most recent and comprehensive release in this series. It contains longitudinal records of $223,291$ unique patients, including diagnoses coded using both ICD-9 and ICD-10 systems. The sequences of ICD codes per patient vary substantially in length, ranging from 1 to $2,396$ codes (median is 13), while the number of admissions per patient ranges from 1 to 238 (median is 1).

**Data preprocessing** In this work, we focus exclusively on the sequences of ICD-10 codes assigned to each patient, as these form the core representation for our modeling and evaluation. We denote the set of patients in the following way:

$$P = \left\{ p_i : \left[ ICD_{p_i}^1, \ldots, ICD_{p_i}^{N_{p_i}} \right], i \in [1, N] \right\}, \tag{1}$$

where $N$ is the number of patients, $p_i$ is the $i^{\text{th}}$ patient with $N_{p_i}$ number of ICD codes.

The patients' sequences preprocessing includes the following steps:

1. Converting all ICD-9 codes to the ICD-10 system using the General Equivalence Mapping[7].

2. Truncating all ICD-10 codes to their 3-character category level, disregarding subcategory details. For instance, the code `C50.911`, representing *"Malignant neoplasm of unspecified site of right female breast"*, was reduced to the broader category `C50`, which covers all malignant neoplasms of the breast. This abstraction allow us to reduce dimensionality while retaining clinically meaningful groupings relevant to disease co-occurrence patterns. The resulting set contained $1,754$ unique three-character ICD-10 categories.

3. Eliminating duplicate pairs (PatientID, ICD-10 code) within a single admission.

#### 3.1.2 ICD CODES DATA

**ICD codes** We utilize the same set of $1,754$ unique three-character ICD-10 categories, that are present in MIMIC-IV. In this type of data, we consider each ICD-10 code as a separate object without any sequential structures.

---

[6] `https://physionet.org`

[7] Released by the Centers for Medicare & Medicaid Services organization in 2018, `https://www.cms.gov/medicare/health-plans/medicareadvtgspecratestats/risk-adjustors-items/risk2018`

**ICD codes with their description** We add textual descriptions of the ICD-10 codes from the paragraph above using the Python library `simple-icd-10`. The average length of descriptions is 5 words consisting of on average 40 characters.

## 3.2 METHODS

The methods for revealing interconnections between diseases can be divided into four categories, which we consider in more detail in the following subsections.

### 3.2.1 METHODS FOR REAL DATA

To analyze patterns of co-occurring diagnoses among real patients, we utilize the two methods: (**M1**) baseline statistical approach and (**M2**) masked language modeling (MLM).

We start with statistical baseline methods that quantify how often pairs of ICD-10 diagnostic codes co-occur in the same patient's medical history and assess the strength of these associations, using two approaches: (**M1.1**) Fisher's exact test and (**M1.2**) a Jaccard-based method.

**M1.1 – Fisher's exact test** We compute co-occurrence statistics between ICD codes by counting, for each ordered pair $(i, j)$, the number of patients who received diagnosis $i$ before $j$. We start with building an $N \times N$ co-occurrence matrix, in which entry $(i, j)$ records these ordered counts. After that, we apply Fisher's exact test to each row–column pair and control the false discovery rate (FDR), identifying 138 "disease $\rightarrow$ disease" associations with adjusted p-value $p < 0.05$. The test evaluates whether two categorical variables are independent by examining the $2 \times 2$ contingency table defined for each ordered pair $(i, j)$. The test considers whether the observed co-occurrence count of diagnosis $i$ preceding $j$ is significantly greater than expected under the null hypothesis of independence. We employ the one-sided version of the test ("greater" alternative), which is sensitive to enrichment in the $i \rightarrow j$ direction. Therefore, the obtained odds ratio quantifies the strength of association. The corresponding $p$-values are adjusted using the Benjamini–Hochberg procedure.

**M1.2 – Jaccard co-occurrence matrix** For each patient, we consider all unique ICD-10 codes and compute a Jaccard-like co-occurrence matrix. For each pair of ICD codes $i$ and $j$, we calculate:

$$J_{i,j} = \frac{N_{i,j}}{N_i + N_j - N_{i,j}}, \tag{2}$$

where $N_{i,j}$ is the number of patients diagnosed with both codes $i$ and $j$, $N_i$ and $N_j$ are the number of patients diagnosed with codes $i$ and $j$, correspondingly.

This measure mirrors the classical Jaccard index by normalizing the number of patients with both diagnoses by the total number of patients in either group. Values near 1 indicate frequent co-occurrence, while values near 0 suggest little or no overlap.

**M2 – MLM** We apply this method to determine if the sequential structure of diagnosis codes contains enough information to reveal meaningful disease relationships, expecting to reveal interconnections based on real patient data patterns.

For MIMIC-IV data preprocessing, we use the following procedure:

1. Filter lengths of sequences of patient ICD codes between 5 and 100, following prior work (Placido et al., 2023).

2. Generated masked sequences according to the BERT pretraining strategy (Devlin et al., 2019): in each sequence, 15% of tokens are selected for prediction; of these selected tokens, 80% are replaced with [MASK], 10% with a random token, and 10% are left unchanged. The model was trained with cross-entropy loss on these tokens.

The model architecture consists of an embedding layer (dimension 128) with positional encodings, followed by 3 Transformer encoder layers (8 heads, feed-forward dimension 512, dropout 0.1). The output is produced through a linear layer. We train for up to 100 epochs using AdamW (learning

rate $5 \times 10^{-4}$, weight decay 0.001) with batch size 128, ReduceLROnPlateau scheduling, and early stopping (patience = 5). Linear layers are initialized with Xavier uniform values, and the embedding layer with normal initialization.

Dataset is split into 80% training and 20% test sets. The final model achieves test accuracy of 0.3011 and test loss of 3.6263.

The detailed description of hyperparameter optimization with Optuna is provided in Appendix A.1.

### 3.2.2 PRETRAINED MODELS ON ICD SEQUENCES AND TEXT DESCRIPTIONS

**M3 – Pretrained Med-BERT** Med-BERT (Rasmy et al., 2020) adapts BERT to medical data by treating a patient's ICD-10 code sequence as a "sentence." Pretrained on large-scale EHRs, it reflects real-world clinical patterns. We use it to obtain ICD-10 code embeddings and compute their full pairwise cosine similarity matrix to capture semantic relationships between diagnoses.

**M4 – Pretrained BioClinicalBERT** We use BioClinical BERT[8] – initialized from BioBERT-Base v1.0 (pretrained on PubMed and PMC) and further trained on all MIMIC-III clinical notes (Huang et al., 2019). Each ICD code is mapped to its long-form description (e.g., "Malignant neoplasm of bronchus and lung"), tokenized, padded/truncated, and passed through the model in evaluation mode. We extract the final-layer [CLS] embedding, L2-normalize all embeddings, and compute the full pairwise cosine similarity matrix.

### 3.2.3 METHODS FOR TEXTUAL DESCRIPTIONS

We utilize the Python library `simple-icd-10`[9] to obtain textual descriptions of ICD codes.

**M5 – Pretrained BERT** We use `bert-base-uncased`, a BERT model pretrained on general (non-medical) text, to embed ICD code descriptions and compute pairwise cosine similarities. It serves as a baseline capturing purely textual similarity without domain-specific medical knowledge.

**M6 – Yandex Doc Search** We employ Yandex Cloud's pretrained `text-search-doc` embedding model – designed for document-level semantic retrieval – to generate semantic representations of ICD descriptions and compute pairwise cosine similarities, providing an additional text-based baseline for identifying disease interconnections.

### 3.2.4 LLM-BASED METHODS

We use LLMs to leverage their pretrained medical knowledge for predicting ICD-10 code co-occurrence patterns; prompt engineering details are in Appendix A.2. The final prompt is the following:

> I'll give you ICD-10 categories (for example, C25, NOT C25.0!) and their descriptions. You have to tell me, If a patient has an ICD code for a given category in their medical record, what other categories of codes are also likely to be in their medical record?
> ANSWER IN JSON FORMAT: { "comment": your thoughts and explanations , "answer": list of categories in square brackets, separated by comma, for example: [A01, C05, ..., H12] } DO NOT ADD ANYTHING ELSE IN YOUR ANSWER.
> TEMPLATE_MULTI = {{ icd_code: {}, description: {}, }}

This prompt is constructed in such a way as to obtain multiple connections for each ICD-10 category and avoid $O(N^2)$ API requests to the model due to the high estimated execution time in such a case (more than three weeks) and prohibitive cost. We test three models using API: **DeepSeek-V3**, **Yandex-GPT 5**, and **Qwen 3-235B-A22B**. The ablation study on the minimum required model's size (including smaller models) is presented in Appendix A.3.

---

[8] https://huggingface.co/emilyalsentzer/Bio_ClinicalBERT
[9] https://pypi.org/project/simple-icd-10/

Table 1: Spearman correlations (value [95% CI]) between LLM and other methods.

| | Basic statistics | | MLM | Text data | | Pretrained on medical data | |
| | M1.1 | M1.2 | | Pretrained BERT | Yandex Doc Search | Pretrained Med-BERT | Pretrained BioClinicalBERT |
|---|---|---|---|---|---|---|---|
| DeepSeek-V3 | 0.079 [0.076, 0.081] | -0.008 [-0.010, -0.007] | 0.115 [0.112, 0.117] | -0.034 [-0.035, -0.033] | 0.107 [0.106, 0.108] | 0.060 [0.058, 0.063] | 0.061 [0.060, 0.062] |
| Yandex-GPT 5 | 0.092 [0.090, 0.095] | 0.054 [0.052, 0.055] | 0.069 [0.066, 0.071] | 0.057 [0.055, 0.058] | 0.093 [0.092, 0.094] | 0.071 [0.068, 0.073] | 0.064 [0.063, 0.065] |
| Qwen 3-235B-A22B | 0.079 [0.076, 0.081] | 0.060 [0.058, 0.061] | 0.045 [0.042, 0.048] | 0.029 [0.028, 0.030] | 0.093 [0.092, 0.094] | 0.049 [0.046, 0.051] | 0.049 [0.048, 0.050] |

## 4 EXPERIMENTS AND RESULTS

### 4.1 OBTAINING DISEASES INTERCONNECTIONS

The disease interconnections obtained from all methods are presented in the Appendix A.4.

Figures 2 and 3 show that MLM and Med-BERT yield similar patterns, whereas BioClinicalBERT produces largely opposite results – high similarity where the others are low, and vice versa. Baseline methods (Fisher's exact test and Jaccard similarity) show some resemblance but generally weak disease pair connections.

Text-based methods (Figure 4) exhibit uniformly high similarity scores, likely due to shared terminology among diseases in the same chapter.

Among LLMs (Figure 5), DeepSeek reveals more interconnections than Qwen and YandexGPT, yet all align with patterns from Med-BERT, MLM, and Jaccard co-occurrence (Figures 3 and 2).

Boxplots (Figures 6–9) show text-based and LLM approaches yield high scores (avg. $\approx 0.8$), while statistical methods and some LLMs cluster near $0$. Medical-domain pretrained models fall in between: Med-BERT averages $0.6$ and BioClinicalBERT $0.2$ (Figure 7).

The following sections offer a detailed qualitative and quantitative comparison of these methods.

### 4.2 COMPARISON OF OBTAINED DISEASES INTERCONNECTIONS

Section 4.1 presents a visual comparison of the obtained disease interconnections using heatmaps. However, visual comparison becomes challenging due to the substantial number of ICD codes, as we generated 10 matrices, each with dimensions of $1646 \times 1646$. Furthermore, traditional matrix comparison methods, such as distance-based and correlation-based approaches, are inappropriate for our analysis since we lack ground truth data. Instead, our objective is to identify similar patterns across groups of methods, which can then be investigated further and selected as the most significant interconnections that appear consistently across nearly all methods.

In the following subsections, we present additional comparative methods for analyzing the obtained disease interconnection matrices.

#### 4.2.1 INTERCONNECTIONS' CORRELATIONS ACROSS DIFFERENT METHODS

After applying the previously described methods to the data, we receive ten disease interconnection matrices. We evaluate them by using pairwise correlations with 95% confidence intervals. We use Spearman correlation used bootstrap resampling (500 iterations) due to its rank-based nature. Matrix alignment was achieved by intersecting common rows and columns before vectorization.

The correlation analysis shows relatively low associations between LLM-generated matrices and other methods (Table 1). All Spearman correlations remain below 0.12, indicating weak effect sizes. However, their 95% confidence intervals exclude zero, confirming statistical significance. DeepSeek-V3 shows moderate correlation with MLM ($p = 0.115$) while demonstrating significant negative correlations with basic statistics (M1.2: $p = -0.008$) and text-based approaches. Yandex-GPT achieves the most consistent performance across methods, with statistically significant correlations ranging from 0.054 to 0.093. Notably, all three LLMs show similarly weak but statistically significant correlations with medically pretrained models (0.049-0.071), suggesting they capture fundamentally different relationship patterns from domain-specific approaches. These results show that LLMs have limited utility for discovering novel disease interconnections, as they produce relationship patterns that reach statistical significance but correlate weakly with all tested models.

### 4.2.2 T-SNE VISUALIZATION

We visualized ICD code embeddings from BERT, Med-BERT, Yandex Doc Search, and MLM models using 2D t-SNE (Figure 11), with each point representing a disease category color-coded by ICD chapter. Med-BERT shows the clearest clustering, with well-separated groups closely matching ICD chapters despite using only ICD codes (no text). Yandex Doc Search and BERT yield moderately coherent clusters – diseases from the same chapter are generally proximate but with less distinct boundaries. MLM embeddings exhibit the weakest separation, though the non-random distribution suggests some learned semantic structure.

### 4.2.3 GRAPH-BASED COMPARISON OF OBTAINED INTERCONNECTIONS

To convert our matrices of disease interconnections to undirected graphs, we calculate for each matrix its 0.95-quantile and consider two ICD codes as connected if their interconnection score is higher than 0.95-quantile. In this procedure, we, on the one hand, lose a lot of interconnections, on the other hand, consider only those of them, in which models are confident. The detailed graphs' characteristics are presented in Appendix A.6.

**Number of "degrees" of ICD codes from the largest connected components**    For each graph, we find the largest connected component. After that, for each vertex-ICD code we calculate its degree – the number of connected other vertices. We obtain the degrees for all methods and for groups of methods. When we group some of our methods, we calculate the "intersection" of their largest components: we consider all ICD codes and edges between them that are present in the graphs of all methods within each group. Figures 12, 13, 14, 15, 16, 17, and 18 show that methods in groups perform differently and have different patterns in graph connections (when comparing their upper parts to the bottom ones). The detailed description of the cancer-related and non-cancer-related interconnections is presented in Appendix A.6.

**PR AUCs without ground truth**    We define three "ground truth indicators": (1) Fisher's exact test as a real-data statistical baseline; (2) pretrained BERT and Yandex Doc Search as text-similarity–based methods; and (3) Med-BERT, trained on real patient ICD sequences. We exclude Jaccard similarity, because it can inflate co-occurrence scores for rare diseases (e.g., yielding $\frac{1}{3}$ similarity from just three (out of huge number of) patients with $[ICD_1]$, $[ICD_2]$, and $[ICD_1, ICD_2]$), and BioClinicalBERT, which produces fewer interconnections (Section 4.2.3).

When Fisher's exact test is the ground truth (Figure 19), Jaccard achieves the highest PR AUC, followed by Qwen (0.055), Med-BERT and BERT (0.053), YandexGPT and Yandex Doc Search (0.051), and DeepSeek (0.050). Notably, Med-BERT performs similarly to non-medical BERT.

When textual methods serve as ground truth (Figure 20), pretrained BERT and Yandex Doc Search show strong mutual alignment (PR AUCs 0.124 and 0.131). Med-BERT aligns more closely with Yandex Doc Search (0.174) than with BERT (0.069), suggesting Yandex Doc Search may incorporate medical text. Figure 21 confirms the superiority of Yandex Doc Search relatively to Med-BERT with the highest PR AUC 0.182.

We formulate the following hypotheses:

- **H1:** Med-BERT does not rely purely on textual data. To test this hypothesis, we examine cases that are present in pretrained BERT but absent in Med-BERT and assess their semantic similarity.

- **H2:** (**H2.1**) MIMIC-IV contains noisy data or (**H2.2**) these represent previously unknown disease interconnections. To investigate this, we examine in detail (**H2.1**) interconnections that are present in Fisher's exact test but absent in Med-BERT and, conversely, (**H2.2**) interconnections that are present in Med-BERT but absent in Fisher's exact test. (**H2.1**).

Since Yandex Doc Search and YandexGPT are the models most similar to Med-BERT according to Figure 21, we include them alongside Med-BERT in our subsequent experiments.

**H1: Med-BERT does not purely rely on textual data**    We identify 118492 ICD code pairs interconnected by pretrained BERT but not by Med-BERT. Grouping codes by chapter, we compute

semantic similarities for all intra-chapter pairs using both plain ICD descriptions and descriptions prefixed with ICD codes, yielding mean similarities of $0.534\pm0.118$ and $0.616\pm0.089$, respectively. If BERT relied solely on textual descriptions, its similarities would align with these baselines. However, Figure 10 (left) shows most similarities fall in the 0.2–0.3 range, indicating pretrained BERT leverages non-textual medical information – confirmed by examining its pretraining data:

- **English Wikipedia:** contains extensive health- and medicine-related content, attracting billions of annual page views and linking to substantial academic research. Dedicated initiatives assess and improve its medical articles, with studies examining their readability and popularity (Farič et al., 2024; Brezar & Heilman, 2019). Specialized corpora like the "Wikipedia Human Medicine Corpus" (Marcos et al., 2017) and "Wiki[Alt]Med corpus" (Jones, 2025) are built directly from this content.
- **BookCorpus (Zhu et al., 2015):** composed mainly of unpublished fiction and non-fiction by indie authors, may include medical or health-themed books, though this is less systematically documented than in Wikipedia.

We also tested interconnections present in Med-BERT but absent in pretrained BERT. The mean semantic distance for $116482$ such ICD code pairs is $0.531\pm0.121$ for descriptions and $0.614\pm0.092$ for descriptions with ICD codes. As shown in Figure 10 (right), their similarity scores (0.1–0.2) are even lower than in the previous case, indicating that most Med-BERT-derived interconnections exhibit low semantic similarity – lower than pretrained BERT's – suggesting Med-BERT relies on non-semantic forms of similarity.

Both experiments (1) confirm that Med-BERT does not rely solely on textual descriptions and (2) demonstrate that pretrained BERT also possesses some knowledge in the medical domain.

**H2: Noise or surprisingly novel disease interconnections?** We begin by examining ICD code pairs that are statistically associated via Fisher's exact test but not linked by Med-BERT, Yandex Doc Search, and YandexGPT. These form a connected graph of $981$ ICD codes. Focusing on lung cancer (C34: "Malignant neoplasm of bronchus and lung") and prostate cancer (C61: "Malignant neoplasm of prostate"), we find that both share five frequently co-occurring comorbidities among their top 10 most reported codes: E11 (type 2 diabetes mellitus), E78 (disorders of lipoprotein metabolism), I10 (essential hypertension), I25 (chronic ischaemic heart disease), and Z87 (personal history of other diseases).

All five rank among the top-20 most frequently co-occurring ICDs in Fisher's exact test. Their frequent co-occurrence reflects real-world multimorbidity patterns driven by shared risk factors (e.g., age, obesity, sedentary lifestyle) (Tazzeo et al., 2023; Franken et al., 2023) and pathophysiological links – particularly metabolic syndrome (Rus et al., 2023). Clinically, type 2 diabetes (E11) commonly coexists with hypertension (I10) and dyslipidaemia (E78) (Alawdi et al., 2024), forming a triad that markedly increases the risk of chronic ischaemic heart disease (I25) (Al-Ghamdi et al., 2022). Z87 often captures relevant prior events (e.g., stent placement or transient ischaemic attack) that inform ongoing care. Thus, these associations likely represent genuine clinical patterns rather than data artifacts.

Accordingly, we cannot conclude that MIMIC-IV contains noisy data – though we currently lack definitive evidence either way. All results are available in our GitHub repository.

We also analyze the complementary set: ICD pairs linked by Med-BERT, Yandex Doc Search, and YandexGPT but not by Fisher's exact test. This yields a large connected component of 1600 ICD codes. For C34, our models identify the following top-associated conditions: malignant neoplasm of stomach (C16), other and ill-defined digestive organs (C26), larynx (C32), trachea (C33), thymus (C37), heart, mediastinum and pleura (C38), peripheral nerves and autonomic nervous system (C47), retroperitoneum and peritoneum (C48), adrenal gland (C74); C45 (mesothelioma), D02 (carcinoma in situ of respiratory system), D15 (benign intrathoracic neoplasm), D38 (neoplasm of uncertain behaviour in respiratory/intrathoracic sites), and E10 (type 1 diabetes mellitus). Several of these have plausible clinical explanations: C33 and C38 can be results of anatomical contiguity or invasion (Al-Ayoubi & Flores, 2016), C74 and C48 – metastatic spread (Bazhenova et al., 2014; Nishiyama et al., 2016), C37 and C47 – paraneoplastic syndromes (Hernández et al., 2021), D02 and D38 – pathological progression (Gardiner et al., 2014; Lambe et al., 2020), D15 – diagnostic mimicry (Homrich et al., 2015). The remaining codes – C16, C26, C32, C45, and E10 – lack clear

mechanistic explanations in current literature and may represent underexplored or novel disease interconnections.

**Diseases, connected with top-10 cancers according frequency in MIMIC-IV**    Sections 4.2.3 and 4.2.3 show that even text-based models have some medical knowledge, that is why we also consider them alongside with the others.

Table 3 presents the top-10 cancers according to frequency in our data. We choose non-secondary neoplasms with certain behavior, and provide plots for all∗ of them in our repository. In this paper, Figures 22 and 23 show the radar plots for cancer-related and non-cancer-related ICDs connected with C34, respectively. All plots consider all 10 models: (real data-based) Fisher exact test, Jaccard similarity, MLM; (pretrained models on medical domain) pretrained Med-BERT, pretrained Bio-ClinicalBERT; (text-based approaches) pretrained BERT, Yandex Doc Search; and (LLMs) Qwen-3, DeepSeek-v3, YandexGPT-5. We can see that most of models (five or more) indicate C47, C38, C74, C37, C33, C32, C45, C16 as the connected ones with C34 (they are also considered in the previous paragraph). Four models indicate C26 and C48 as connected with C34 ; three models – D02, D38, two models – E10. Two models identified D15 as connected with C34; however, D15 during plotting was categorized as non-cancer.

## 5    CONCLUSION AND DISCUSSION

We derived ICD-10 disease interconnections from patient co-occurrence data and compared multiple methods: (1) real-data approaches (Fisher's exact test, Jaccard similarity, MLM); (2) medical-domain models (Med-BERT, BioClinicalBERT); (3) general text models (BERT, Yandex Doc Search); and (4) LLMs (DeepSeek, Qwen, YandexGPT). Methods within the same category did not consistently yield similar interconnection patterns.

Lacking ground truth, we introduced an unsupervised evaluation procedure. Results show even general text models encode medical knowledge, making them viable complements to EHR-based and LLM approaches for studying disease relationships.

Manual inspection of MIMIC-IV revealed no clear noise, though we cannot confirm the data is entirely clean due to limited sampling analysis. All interconnections, visualizations, graphs, and the ontology are publicly available on GitHub.

All the findings mentioned above outline the future directions of our research. First, we plan to investigate the presence of noise in the MIMIC dataset in more detail. Second, we will examine how the identified interconnections and model embeddings affect downstream tasks, such as cancer prediction and mortality risk assessment. Finally, we aim to incorporate an unsupervised quality assessment of LLMs.

## 6    LIMITATIONS

Our analysis uses MIMIC-IV, an ICU-only dataset that excludes healthy individuals and may not reflect the general population.

Diagnosis sequence construction involves nuances: chronic conditions (e.g., E06.3) are often repeatedly coded regardless of further admissions. Additionally, ICD-10 coding can introduce dependencies, e.g., E11.65 (type 2 diabetes with hyperglycemia) inherently includes hyperglycemia, making a separate R73.9 code redundant, though R73.9 can occur independently in other contexts.

These coding complexities were not explicitly modeled, and we did not assess whether the evaluated models learned such relationships during training.

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

Table 2: Hyperparameter search space for MLM training

| HYPERPARAMETER | SEARCH SPACE |
|---|---|
| Learning rate | $\{5 \times 10^{-5}, 1 \times 10^{-4}, 3 \times 10^{-4}, 5 \times 10^{-4}, 1 \times 10^{-3}\}$ |
| Weight decay | $\{1 \times 10^{-3}, 1 \times 10^{-2}\}$ |
| Dropout | $\{0.05, 0.1, 0.2\}$ |
| Embedding dimension | $\{128, 256, 512\}$ |
| Number of Transformer layers | $\{1, \ldots, 7\}$ |
| Number of attention heads | $\{1, \ldots, 8\}$ |
| Feed-forward hidden dimension | $\{256, 512, 1024\}$ |
| Output layer depth | $\{1, 2, 3\}$ |

## A APPENDIX

### A.1 TECHNICAL DETAILS OF MLM TRAINING

We performed hyperparameter optimization using Optuna with Median Pruner to accelerate trial selection. The objective function minimized validation loss. Patients were randomly split into training (80%) and validation (20%) sets at the subject level to prevent leakage. The hyperparameter search space is presented in Table2. For the output projection, we experimented with three alternatives:

1. a single linear layer,

2. two linear layers with GELU and dropout in between,

3. three linear layers with GELU activations and dropout between each layer.

Each trial was trained for up to 100 epochs with batch size 128, AdamW optimizer, and ReduceLROnPlateau scheduler. Early stopping (patience = 3) was applied to reduce overfitting. All experiments were conducted on a single NVIDIA A100 GPU (40GB VRAM).

### A.2 PROMPT ENGINEERING FOR OBTAINING INTERCONNECTIONS VIA LLMS

After several additional experiments, we set a requirement: the model must return results strictly in JSON format with two mandatory fields – comment (a meaningful analysis of the relations) and answer (a clear list of categories in square brackets). The comment field was included to make sure the model would not skip the requirement of generating the ICD code list while providing its reasoning. In addition, we introduced a restriction against any extra information outside the JSON structure. This helped minimize variability and ensure reproducibility of the results.

### A.3 THE NECESSARY MINIMUM OF LLMS' PARAMETERS FOR ITS REASONABLE BEHAVIOR

For the LLM research block, we used large-scale models such as Qwen3-235B-A22B, YandexGPT-5, and DeepSeek-V3. However, during the study, an important question arose: at what parameter scale do language models begin to produce meaningful answers, and can we define an approximate threshold for this transition?

To explore this, we conducted experiments with a wide range of models, focusing mainly on Mistral-7B and the DeepSeek-R1-Distill-Qwen family (1.5B, 7B, 14B, 32B). When tested with the established prompt, Mistral-7B often returned a list of all ICD codes it recognized or produced other irrelevant outputs. More stable behavior was observed in DeepSeek-R1-Distill-Qwen (1.5B and 7B), as well as in Granite-3.2 (8B). These models did not generate completely nonsensical results, but they consistently failed to follow the required unified JSON format, which we attribute to the limited parameter size.

With DeepSeek-R1-Distill-Qwen (14B), we achieved greater stability in keeping the JSON structure, although the formatting varied and required complex post-processing. In contrast, the 32B variant of the same family was able to meet the prompt requirements reliably.

Still, assessing the meaningfulness of the answers remains difficult. To approximate this, we calculated the MSE metric, using as ground truth the relation matrix produced by Qwen3-235B-A22B.

For the DeepSeek-R1-Distill-Qwen models, we expected the mean squared error (MSE) between their learned interconnections and those of the original model to decrease as the number of parameters increases. However, Figure 1 reveals a slightly different trend. This discrepancy can be attributed to two factors: (1) the MSEs were computed using only 101 interconnection distance (ICD) measurements, rather than the full set of 1646; and (2) the DeepSeek-R1-Distill-Qwen models were fine-tuned from the Qwen2.5 series, whereas our reference model is Qwen-3-235B-A22B.

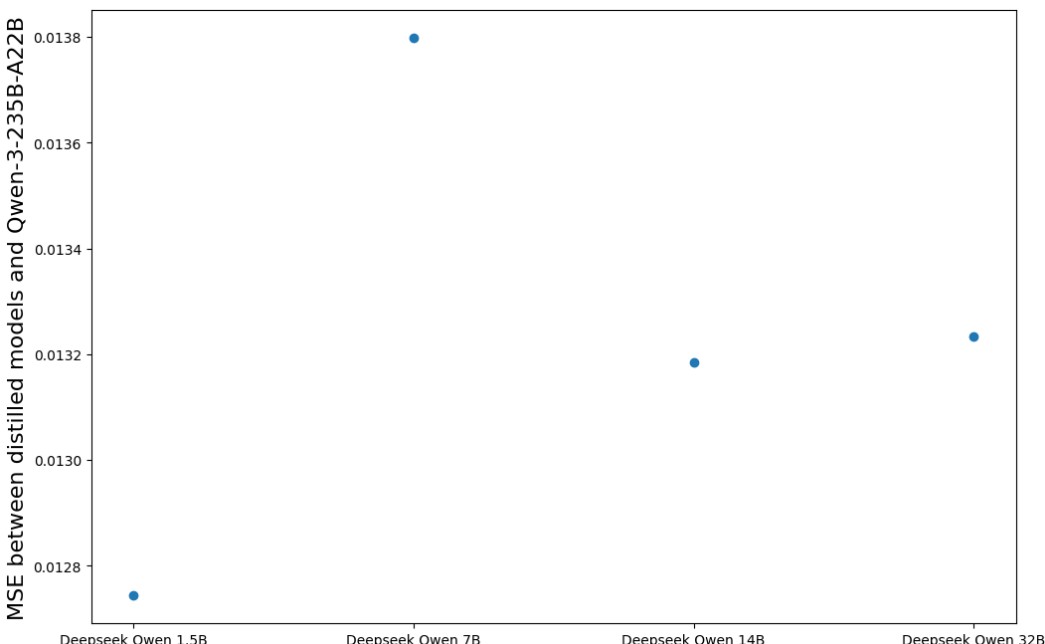

Figure 1: Mean squared error (MSE) between the original Qwen model and its DeepSeek distillations.

### A.4 VISUALIZATION OF MATRICES OF DISEASES INTERCONNECTIONS

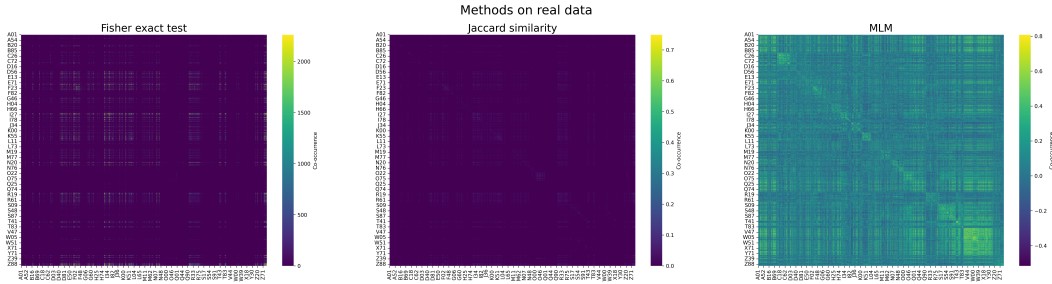

Figure 2: Disease interconnection matrices of methods working with real data: statistical approaches (Fisher's exact test and Jaccard similarity) and MLM. For Fisher's exact test, we substitute all elements higher than 0.997-quantile as 0.997-quantile, which equals to 2262. This technique is implemented as the number of co-occurrences for Fisher's exact test varies from 0 to 91108.

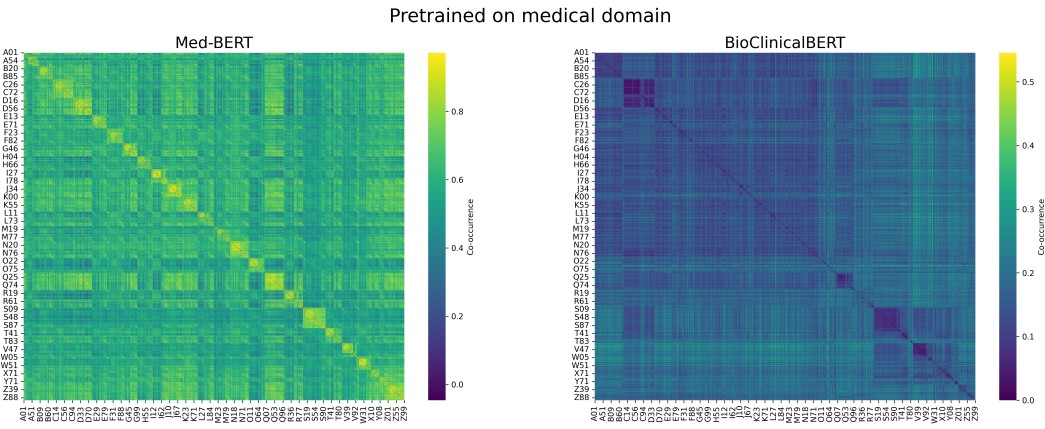

Figure 3: Disease interconnection matrices of methods pretrained on medical domain data: Med-BERT and BioClinicalBERT.

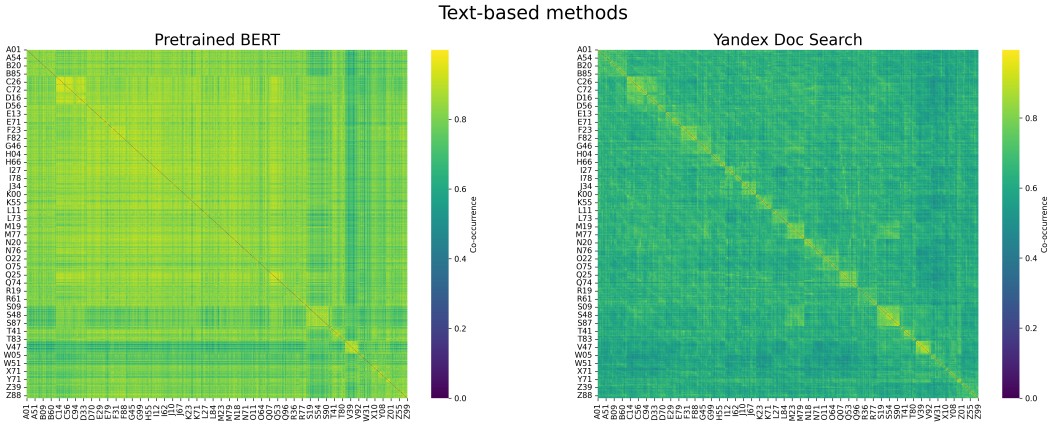

Figure 4: Disease interconnection matrices of methods working with ICD codes' textual descriptions: pretrainde BERT and Yandex Doc Search.

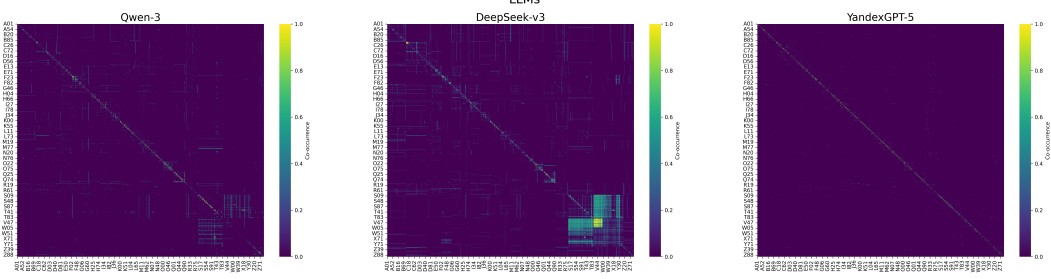

Figure 5: Disease interconnection matrices of LLMs.

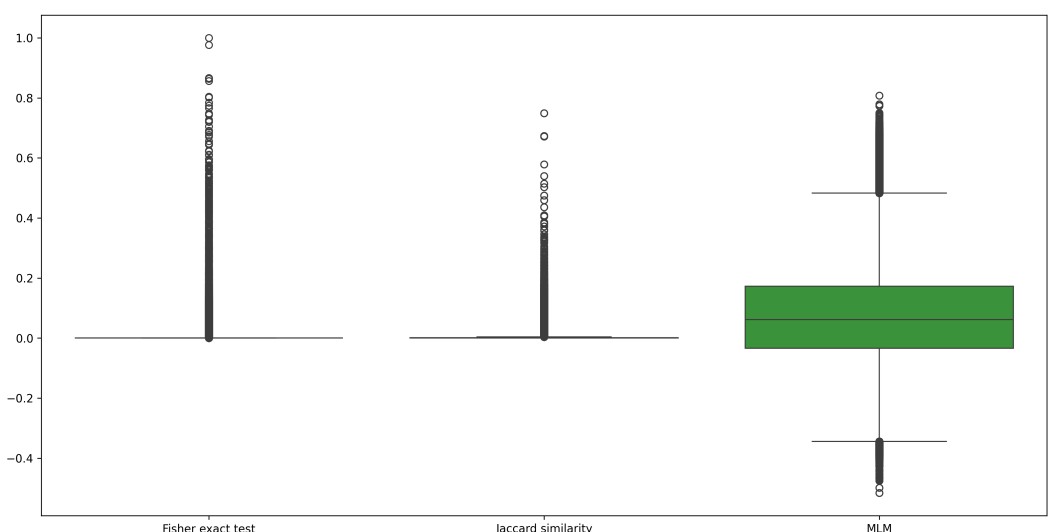

Figure 6: Boxplots of disease interconnection scores of methods working with real data: statistical approaches (Fisher's exact test and Jaccard similarity) and MLM. For Fisher's exact test, we normalize all scores using MinMax-scaling.

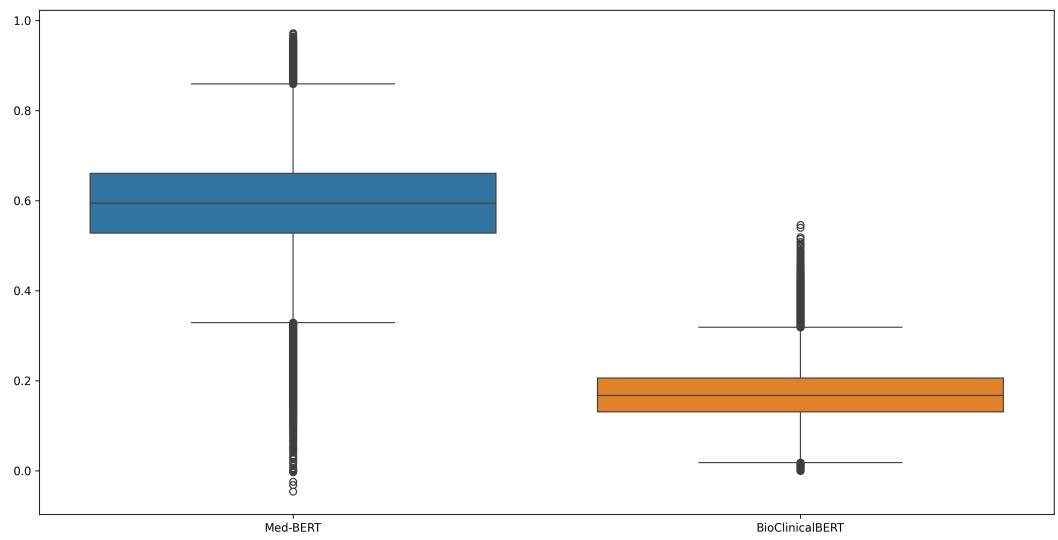

Figure 7: Boxplots of disease interconnection scores of methods pretrained on medical domain data: Med-BERT and BioClinicalBERT.

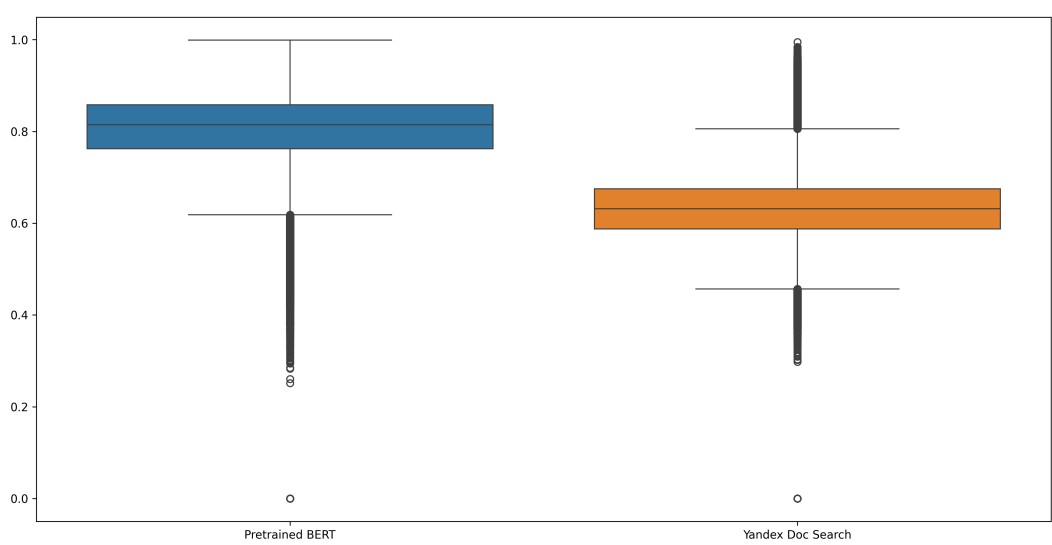

Figure 8: Boxplots of disease interconnection scores of methods working with ICD codes' textual descriptions: pretrainde BERT and Yandex Doc Search.

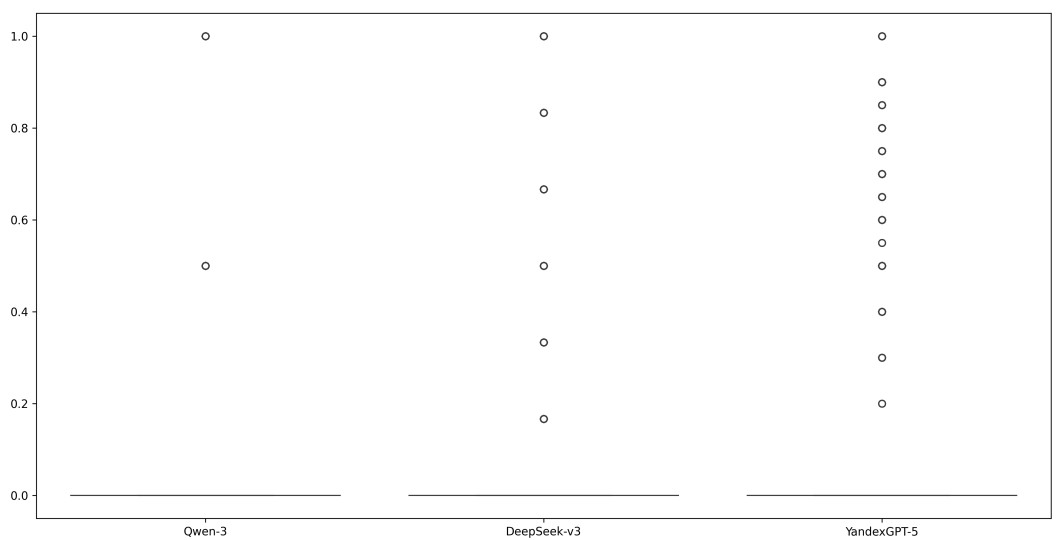

Figure 9: Boxplots of disease interconnection scores of LLMs.

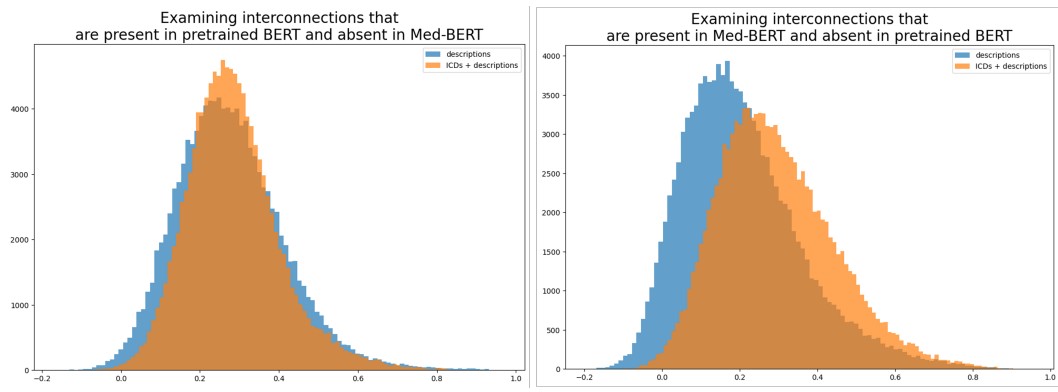

Figure 10: Results for testing **H1**: distributions of similarities between ICD pairs. Left: pairs, which are present in pretrained BERT and absent in Med-BERT. Right: pairs, which are present in Med-BERT and absent in pretrained BERT.

## A.5    T-SNE VISUALIZATION DETAILS

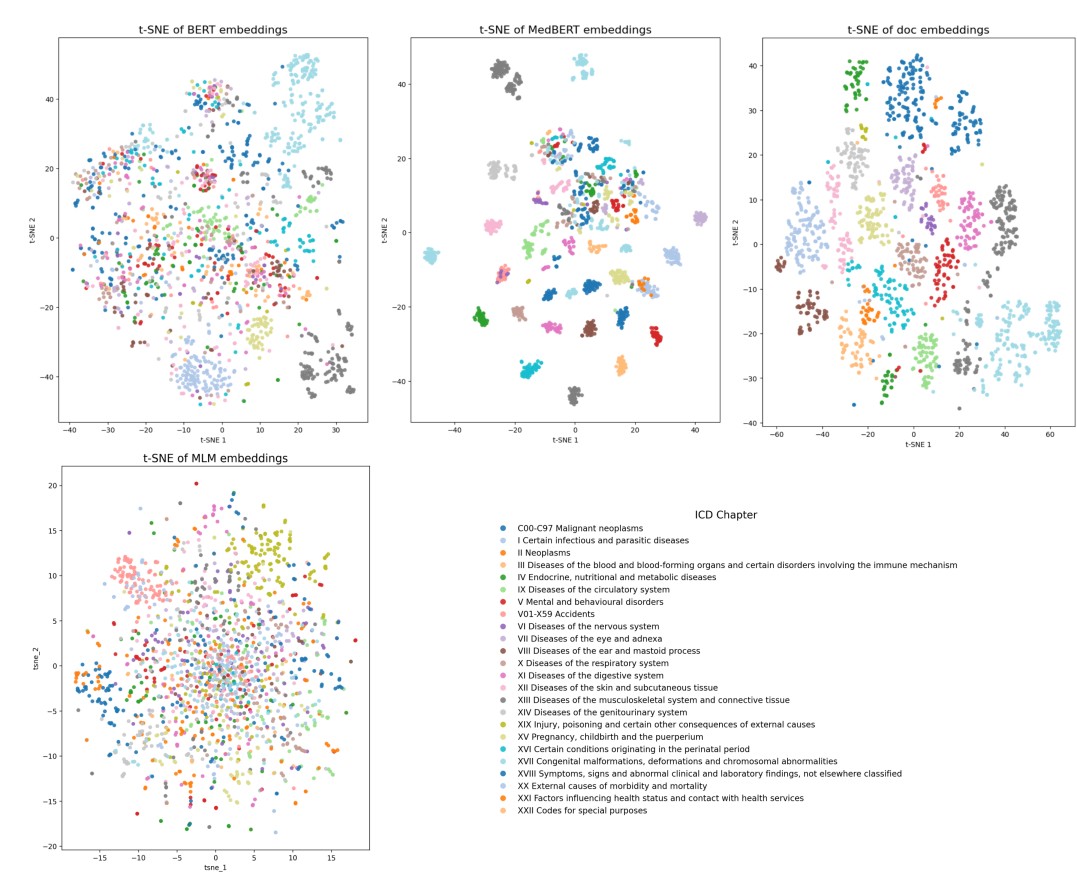

Figure 11: T-SNE visualization of methods where embeddings are produced.

A.6    OBTAINED GRAPHS' DEGREES

This Section presents the more detailed analysis (mentioned in Section 4.2.3) of the "degrees" of obtained graphs.

Figure 12 shows that Fisher's exact test, pretrained BERT, and Jaccard/MLM/DeepSeek have the widest variety in the number of connections identified, suggesting these approaches are capable of detecting diverse relationship patterns across the disease spectrum. Moreover, Fisher's exact test, Yandex Doc Search, pretrained Med-BERT, DeepSeek, and pretrained BERT showed the highest mean number of connections, indicating these methods tend to identify more interconnected disease networks overall. Notably, YandexGPT and Qwen exhibited the narrowest ranges with values close to zero, suggesting these LLM-based approaches may be more conservative in establishing disease connections or may require different optimization strategies for medical domain tasks.

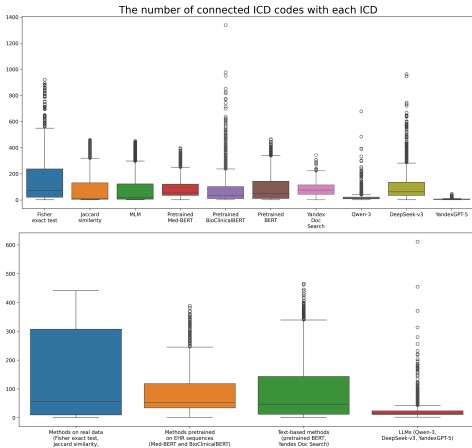

Figure 12: Boxplots of the number of connected diseases for each ICD code. Up: all methods presented separately. Bottom: methods are grouped by the training data type.

When examining connections specifically involving cancer-related ICDs, distinct methodological patterns emerged. Figure 13 shows that Fisher's exact test, Med-BERT, pretrained BERT, and Yandex Doc Search have wide ranges in connectivity patterns, indicating high variability in how these methods assess cancer-related disease relationships. This variability could reflect the complex and heterogeneous nature of cancer pathophysiology and its interactions with other medical conditions. DeepSeek presents a unique profile with small ranges but consistently high values (approaching 100), suggesting this method identifies strong, consistent connections between cancer-related diseases. MLM demonstrates similar patterns to BioClinicalBERT and Jaccard, indicating convergent results among these real data and medical-based approaches for cancer-related connectivity.

The analysis of connections between non-cancer-related diseases (Figure 14) reveals relatively consistent performance across most methods, with some notable exceptions. Fisher's exact test maintaines the highest range of connectivity patterns, while LLMs consistently show the lowest ranges.

Figure 15 shows that Yandex Doc Search, pretrained BERT, and Med-BERT demonstrate the widest ranges and highest averages for cancer-to-cancer connections, indicating these methods are particularly effective at identifying comorbidities and related conditions across cancer categories. The same effect is demonstrated by Figure 16 – these methods capture well the interconnections between cancers and other ICD codes.

Figures 17 and 18 show more or less the same range and average for all methods, except for LLMs. The surprising thing is that DeepSeek performs alongside the other methods, meaning that it better captures non-cancers interconnections.

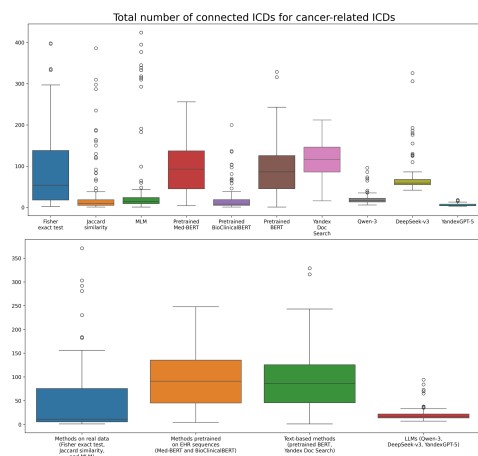

Figure 13: Boxplots of the number of connected diseases for each cancer ICD code. Up: all methods presented separately. Bottom: methods are grouped by the training data type.

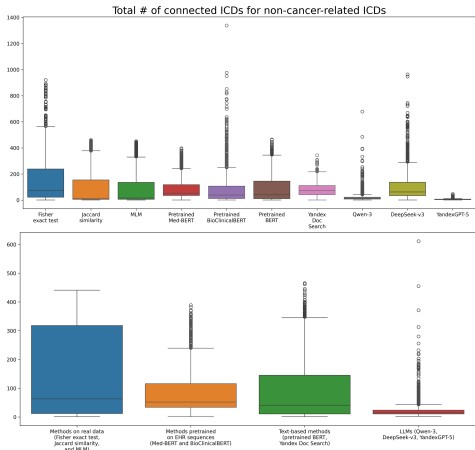

Figure 14: Boxplots of the number of connected diseases for each non-cancer ICD code. Up: all methods presented separately. Bottom: methods are grouped by the training data type.

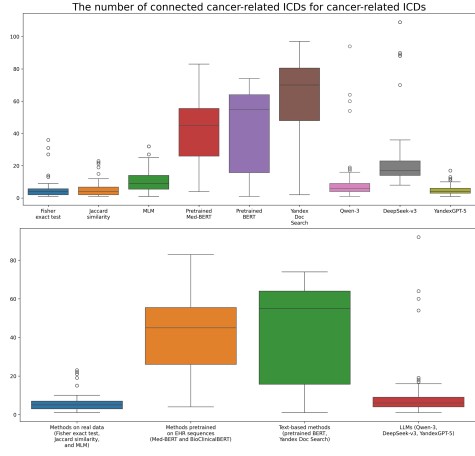

Figure 15: Boxplots of the number of connected cancers for each cancer ICD code. Up: all methods presented separately. Bottom: methods are grouped by the training data type.

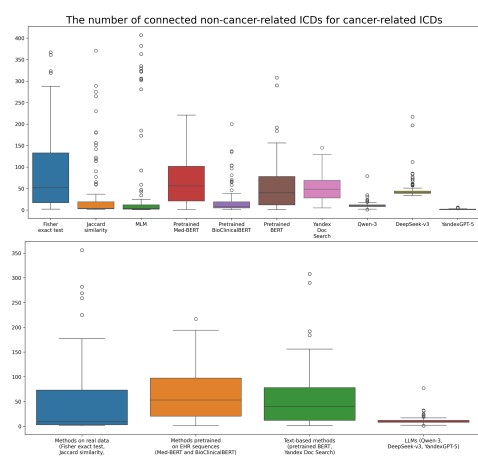

Figure 16: Boxplots of the number of connected non-cancers for each cancer ICD code. Up: all methods presented separately. Bottom: methods are grouped by the training data type.

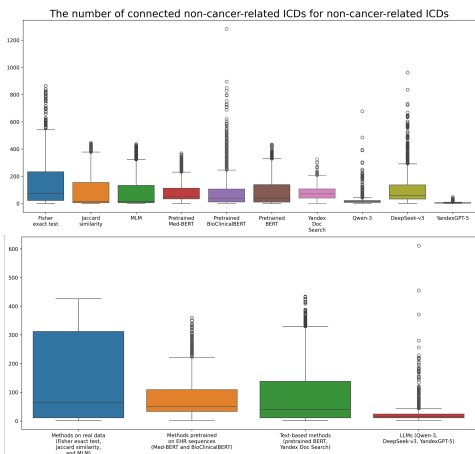

Figure 17: Boxplots of the number of connected non-cancers for each non-cancer ICD code. Up: all methods presented separately. Bottom: methods are grouped by the training data type.

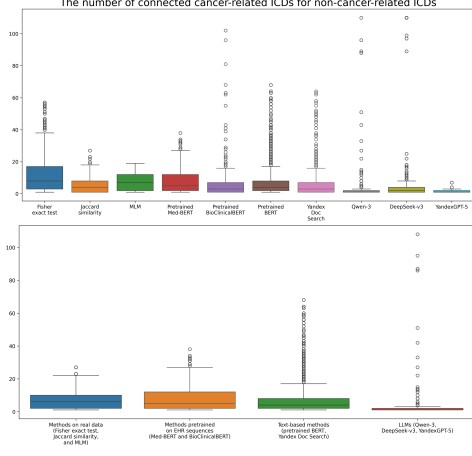

Figure 18: Boxplots of the number of connected cancers for each non-cancer ICD code. Up: all methods presented separately. Bottom: methods are grouped by the training data type.

## A.7 PR AUC SCORES' FIGURES

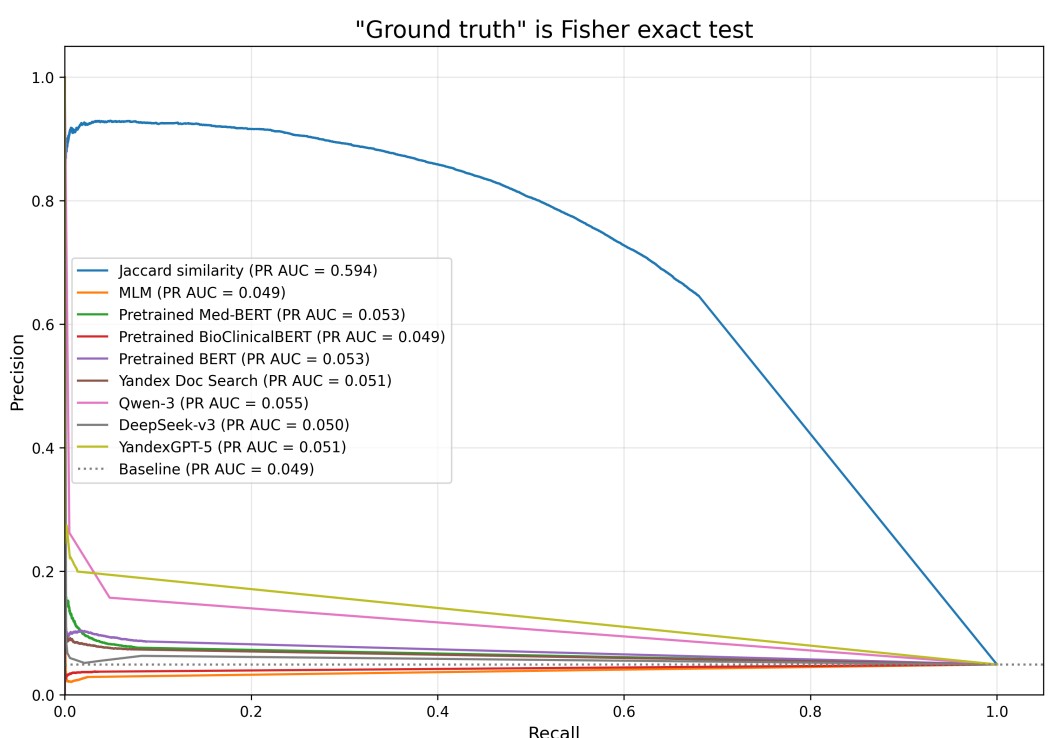

Figure 19: PR AUC scores for all methods, when Fisher's exact test's results are considered as "ground truth".

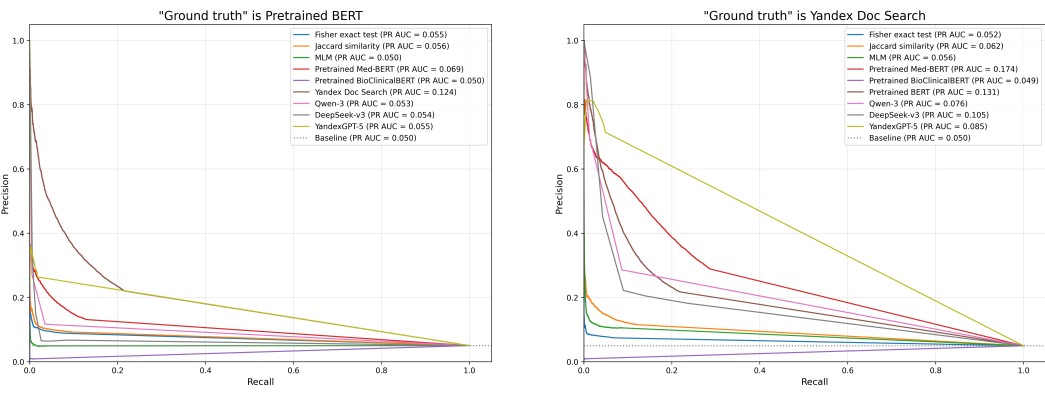

Figure 20: PR AUC scores for all methods, when pretrained BERT's (left) and Yandex Doc Search's (right) results are considered as "ground truth".

Table 3: Top-10 cancers according frequency in MIMIC-IV. The considered ICD codes for visualization are highlighted with ∗

| ICD CODE | NUMBER OF PATIENTS | ICD CODE DESCRIPTION |
|---|---|---|
| C78 | 7724 | Secondary malignant neoplasm of respiratory and digestive organs |
| C79 | 7534 | Secondary malignant neoplasm of other and unspecified sites |
| C77 | 4907 | Secondary and unspecified malignant neoplasm of lymph nodes |
| C34∗ | 4215 | Malignant neoplasm of bronchus and lung |
| D47 | 3554 | Other neoplasms of uncertain or unknown behaviour of lymphoid, haematopoietic and related tissue |
| C61∗ | 2497 | Malignant neoplasm of prostate |
| C50∗ | 2225 | Malignant neoplasm of breast |
| C25∗ | 1958 | Malignant neoplasm of pancreas |
| C22∗ | 1771 | Malignant neoplasm of liver and intrahepatic bile ducts |
| C18∗ | 1403 | Malignant neoplasm of colon |

Figure 21: PR AUC scores for all methods, when Med-BERT's results are considered as "ground truth".

### A.8 RADAR PLOTS OF CANCER- AND NON-CANCER-RELATED ICDs CONNECTED WITH C34

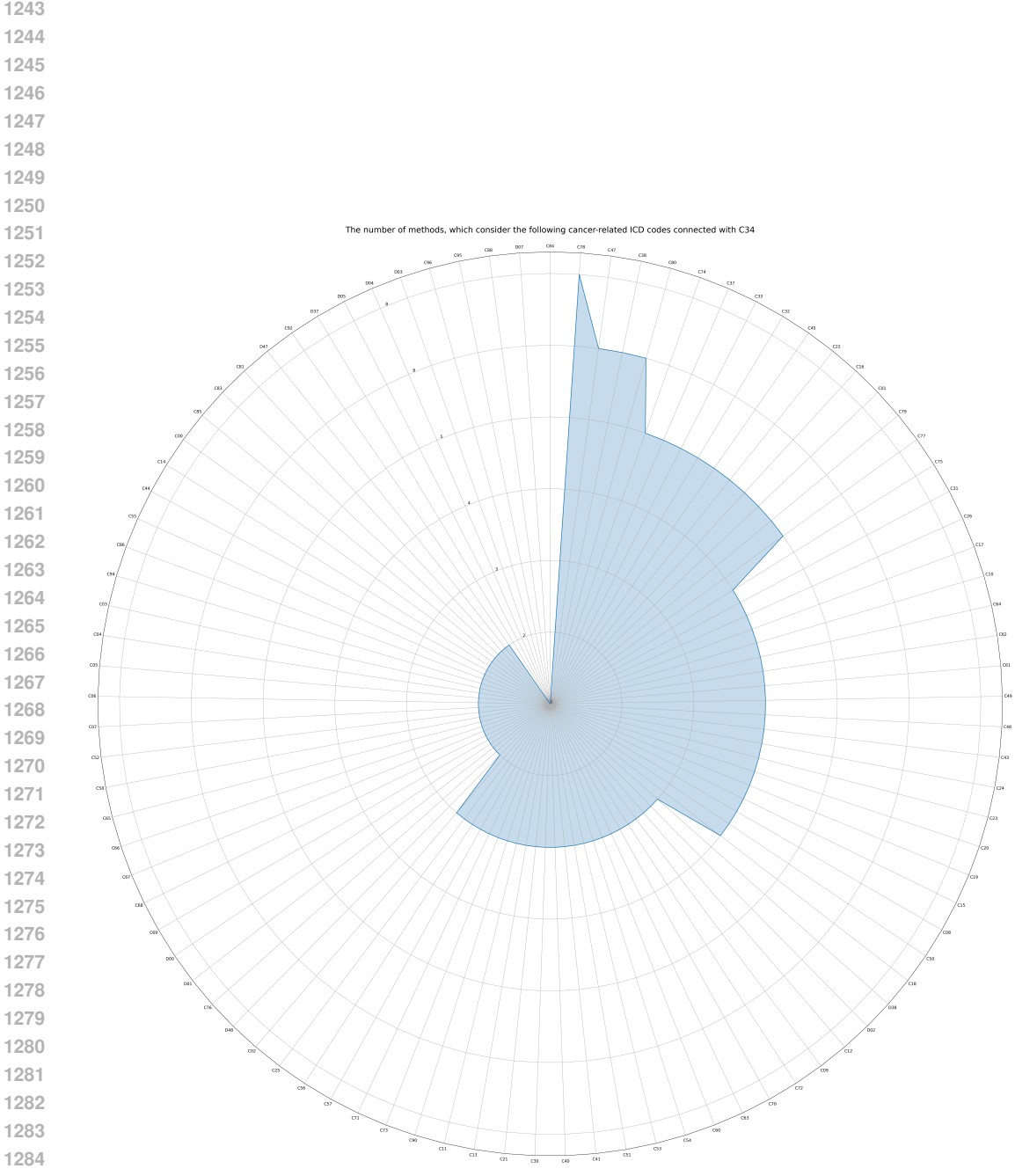

Figure 22: Radar plot of all connected cancer-related ICDs with C34.

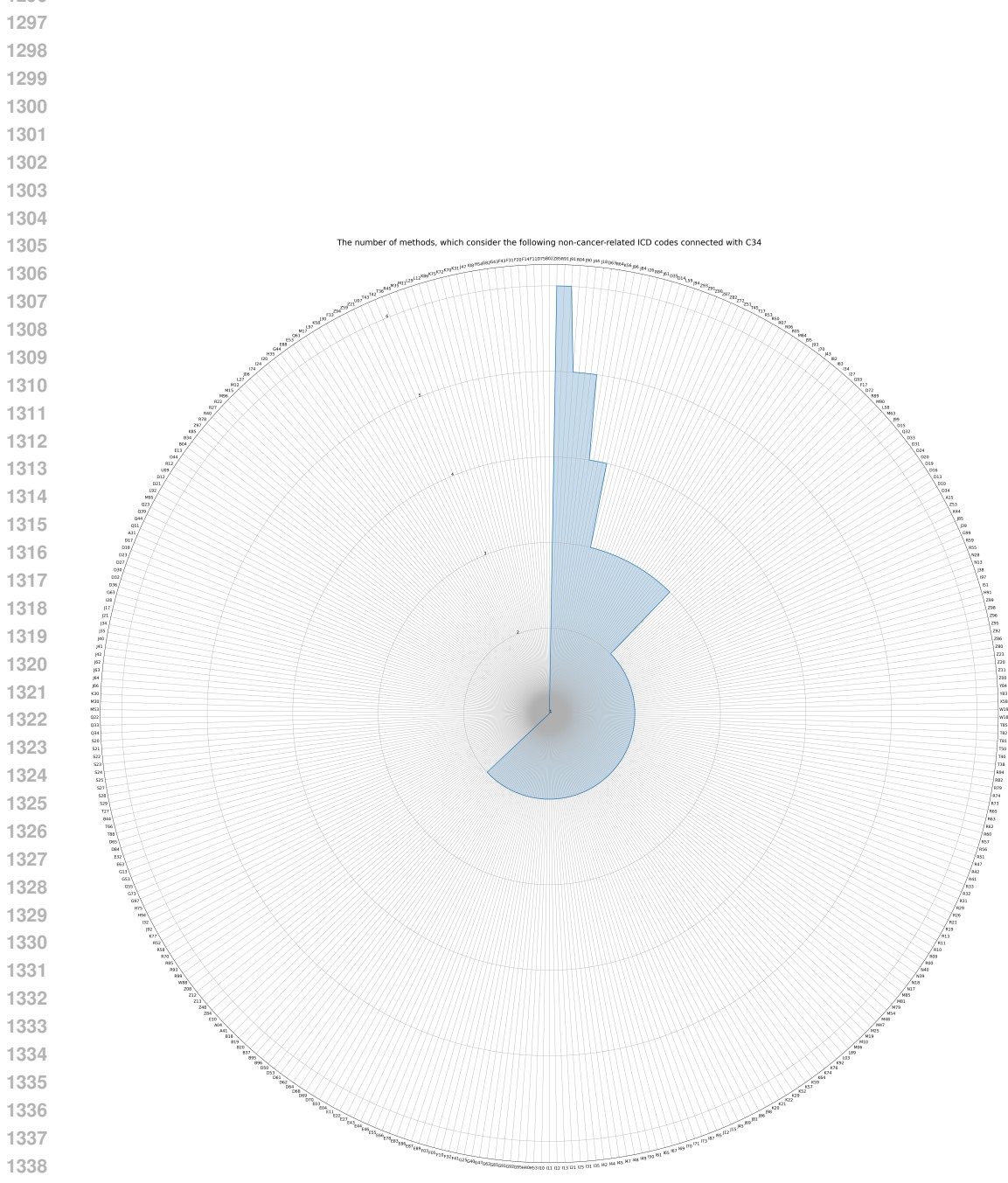

Figure 23: Radar plot of all connected non-cancer-related ICDs with C34.

