# OpenReview forum: "Revealing Interconnections between Diseases: from Statistical Methods to Large Language Models"
_ICLR.cc/2026/Conference — Submitted to ICLR 2026_

### Official Review · Reviewer_SUL2 · 2025-10-22

**Soundness:** 1
**Presentation:** 3
**Contribution:** 2
**Rating:** 2
**Confidence:** 4

**Summary:**

In this paper, the authors use various different techniques to uncover links between ICD10-coded disorders in MIMIC 4's EHR records. They do this by systematically comparing ten methods including two statistical co-occurrence-based metrics (Fisher’s exact test, Jaccard index), a custom masked-language-model (MLM) trained on ICD sequences, two domain-specific Transformers (Med-BERT, BioClinicalBERT), two general text encoders (bert-base-uncased, Yandex Doc Search), and three LLM APIs (DeepSeek-V3, Qwen-3-235B-A22B, YandexGPT-5). They then assess whether these techniques yield similar or dissimilar results, and compare the various methods on their merits. They do this by presenting a suite of unsupervised analyses (Spearman correlations, t-SNE visualizations, graph-based degree statistics, and “PR-AUC” curves under various proxy “ground truths”) and aggregating the interactions into a consensus medical ontology. All code, matrices, and visualizations are released in the supplements.

**Strengths:**

The authors tried different approaches and trained their own models where appropriate, and took a very good approach towards exploring the space; there are without doubt publication-worthy findings in this paper. Their processing of MIMIC IV is interesting, and their discussion of the limitations shows a lot of thoughts went into the writing of this paper. Open-sourcing interconnection matrices, code, hyperparameter logs, and visual assets fosters reproducibility and follow-up work.

**Weaknesses:**

While the paper contains many useful analyses, several key issues limit its impact and warrant deep revision before acceptance. The first is that there is limited algorithmic novelty. The second is that there are methological issues which make the results untrustworthy. The third is that there is insufficient knowledge of the state of the art demonstrated in this work, in my opinion.

The most critical issues:
- Citations of usage of the Med-BERT model (Rasmy, 2020) contrast with the usage (in the provided supplementary materials) of the totally unrelated and rather obscure MedBERT model (Charangan, 2022). Unfortunately, the former model is not available in the open, to my best knowledge, which is why I even bothered to take a look.
- The evaluation of LLMs doesn't use a prompt which looks satisfactory to me. It assume wide knowledge of ICD codes by the models, and tries to evaluate all possible connections from a base concept at once, without providing the LLMs with the option to nuance their predictions. The current LLM matrix is also too sparse to be taken at face value. The argument advanced by the authors that this is done to avoid n^2 LLM calls is relevant, but not convincing in this particular case where only 1.7k concepts categories are considered. The number of pairs is just over a million, and the cost of a million tokens for Deepseek-V3 is $0.42. A well-conceived prompt with a MCQ would unlikely need more than one output token, and you could even get probabilities ("""X is <desc>. Y is <desc>. Based on this, answer the following MCQ: A patient known to have been assigned code X is [A] as likely to have Y than a random patient [B] more likely to have Y than a random patient [C] less likely to have Y than a random patient. Answer with one letter only (A, B or C).""").
- The concept embeddings were computed using models which were not finetuned to produce concept representations. BioClinicalBERT and bert-base-uncased are only trained for masked LM; they are not optimized for semantic similarity. Models like BioLORD or MedCPT (fine-tuned on concept synonym detection) would likely produce more medically meaningful embeddings.

**Questions:**

* Could you clarify which exact Med-BERT checkpoint did you use? Is it publicly available?
* Have you tried medically fine-tuned semantic models (BioLORD, MedCPT)? If so, how do their interconnection matrices compare?
* Can you quantify how much the LLM outputs improve (in terms of consistency, coverage, calibration) when you switch to MCQ‐style pairwise prompts, and evaluate all potential connections?
* How sensitive are your graph‐based degree statistics and consensus ontology to the choice of the top‐quantile cutoff?
* Could you provide a brief stratified analysis showing how common vs. rare ICD codes behave across your ten methods?
* Beyond ontology construction, have you tested whether adding these learned interconnections (e.g., as graph features) improves any predictive tasks (e.g., length of stay, readmission)?

---

### Official Review · Reviewer_1aQp · 2025-10-24

**Soundness:** 2
**Presentation:** 3
**Contribution:** 2
**Rating:** 2
**Confidence:** 3

**Summary:**

This paper conducts a systematic comparative study of ten methods for discovering interconnections between diseases, ranging from traditional statistical analyses (e.g., Fisher’s exact test, Jaccard similarity) to pretrained biomedical models and general large language models. Using real-world EHR data from MIMIC-IV and textual ICD-10 descriptions, the authors construct disease-association matrices and evaluate how well different approaches capture true comorbidity patterns. Results show that LLMs exhibit weak correlations with statistical or biomedical models and tend to generate overly conservative or text-similarity-driven predictions, while Med-BERT produces the most clinically meaningful disease clusters. The study also visualizes disease embeddings and provides a consensus ontology that integrates signals across methods.

**Strengths:**

1. Provides a comprehensive, well-structured comparison across statistical, domain-specific, and general LLM approaches.
2. Uses real clinical data (MIMIC-IV) and ICD-10 hierarchy, grounding the analysis in a realistic medical context.

**Weaknesses:**

1. The authors could have analyzed LLM prompting strategies more deeply (e.g., few-shot or reasoning-chain prompts) to test if reasoning-enhanced LLMs behave differently.
2. The authors don’t test whether minor changes in preprocessing (e.g., ICD granularity, code frequency cutoff, patient sampling) would change the discovered networks.
3. MIMIC-IV data reflect ICU patients, a highly specific population with skewed disease distributions. The authors do not test generalizability to other cohorts, so the results may not hold across medical domains.

**Questions:**

See Weaknesses

---

### Official Review · Reviewer_kvn7 · 2025-10-25

**Soundness:** 3
**Presentation:** 2
**Contribution:** 2
**Rating:** 4
**Confidence:** 3

**Summary:**

This paper systematically evaluated seven methods for identifying disease interrelationships using two data sources: ICD-10 code sequences from MIMIC-IV electronic health records and the complete ICD-10 code set with textual descriptions. The evaluation methods included statistical methods (Fisher's exact test and Jaccard similarity), masked language models (MLMs), pre-trained models in the medical domain (Med-BERT and BioClinicalBERT), general text models (BERT and Yandex Doc Search), and four large language models (Mistral, DeepSeek, Qwen, and YandexGPT). The study preprocessed data from 223,291 patients, truncating codes to three characters, resulting in 1,754 unique ICD-10 categories. The results were compared using correlation analysis, t-SNE visualization, graph analysis, and PR AUC calculation. The key finding was that the LLM produced the lowest diversity of disease interrelationships. The paper constructed a disease ontology based on multi-method consensus and conducted literature validation on selected cases.

**Strengths:**

1. The study design is comprehensive and systematic, comparing ten methods across four categories on a single dataset, providing a valuable benchmark and methodological comparison framework for disease relationship identification.
2. In the absence of ground truth, the study proposes an innovative multi-faceted evaluation strategy, including the use of multiple "proxy ground truths" for PR AUC evaluation, graph topology analysis, and visual comparison. This evaluation framework offers significant methodological contributions.
3. The multi-method consensus-based disease ontology constructed has practical application value, and the strategy of assigning confidence by calculating the number of identical interconnected relationships identified by different methods is reasonable and practical.
4. The study demonstrates good reproducibility, with all code, data, and results publicly available, including detailed hyperparameter search procedures and implementation details.
5. Validation was conducted against medical literature for specific case studies, such as lung cancer, demonstrating the clinical plausibility of the identified disease relationships and strengthening the credibility of the method.

**Weaknesses:**

1. Data representativeness is problematic. MIMIC-IV only includes critically ill ICU patients, a highly specialized population that is not representative of disease distribution in the general population. All identified disease relationships are likely subject to significant selection bias. While the paper addresses this in its limitations, it fails to fully discuss the fundamental impact of this on the validity of the conclusions.
2. The paper lacks true ground truth validation. Using different methods as ground truth for evaluation leads to a circular argument. While literature validation was performed, the sample size was too small and the sample size was too selective to support conclusions about the overall method validity.
3. The core method, MLM, performed extremely poorly, with a test accuracy of only 0.3011 and a high loss of 3.6263. Such low performance suggests that the model may not have learned meaningful patterns, yet the paper still uses it as a key comparison without adequate explanation or improvement.
4. Spearman correlation analysis results show that all correlation coefficients are below 0.12. While statistically significant, the actual effect sizes are extremely small. The paper overemphasizes statistical significance while ignoring practical significance. Such weak correlations fail to support the paper's core conclusion that LLM has limited ability to discover new relationships.
5. The LLM method design has obvious flaws. The prompt design that requires multiple associated diseases to be returned at once may lead to conservative responses. The paper does not explore other prompt strategies such as pairwise comparison. Moreover, the conclusion that the LLM has limited ability to discover new relationships may be due to the method design rather than the inherent limitations of the model.
6. Direct clinical expert involvement and validation are completely lacking. All medical interpretations are based on literature citations, and no professional clinicians have assessed the clinical relevance and credibility of these findings. This is a major flaw in medical AI research.
7. The comparison of methods is fundamentally unfair. Different methods use different data sources and input formats. Some are based on actual patient series, while others are based solely on code description text. Direct comparison of their performance outputs is not comparable.

**Questions:**

1. The MIMIC-IV dataset includes only critically ill ICU patients, a highly specialized population whose disease distribution and comorbidity patterns differ significantly from those of the general population. How do the authors ensure the generalizability of the disease interconnections derived from this data? Can they provide evidence that these findings generalize to non-ICU populations?
2. The paper uses different methods as ground truth for PR AUC evaluation, which is a circular argument. Can the authors obtain or construct an independent validation dataset, such as one using a medical knowledge graph, clinical guidelines, or expert-annotated disease relationships as ground truth for validation?
3. The MLM method achieved a test accuracy of only 0.3011 and a high test loss of 3.6263. These performances suggest that the model may not have learned meaningful disease sequence patterns. Why do the authors continue to use it as an important baseline for comparison? Have they attempted to improve the model architecture or training strategy? If this method is inherently inappropriate, should it be removed from the comparison?
4. Table 1 shows that the Spearman correlation coefficients between all methods are below 0.12. While statistically significant, the actual effect sizes are extremely small. How does this weak correlation support the paper's core conclusion regarding the limited ability of LLM to discover new relationships? Are the authors overly reliant on statistical significance and neglect practical implications?
5. For the LLM method, the authors used a prompt design that requires the return of multiple associated diseases at once. This design may lead the model to give conservative answers. Have the authors tried other prompt strategies, such as pairwise disease relationship prediction or stepwise inference? Could the conclusion that LLM has limited power be due to improper prompt design rather than inherent limitations of the model?
6. The paper completely lacks direct clinical expert participation and validation; all medical interpretations are derived from literature citations. Could the authors have invited clinicians to assess the clinical relevance, plausibility, and practical value of the identified disease relationships? Without expert validation, how can we ensure that these findings are meaningful for clinical practice?
7. Different methods use different data sources and input formats: some are based on actual patient ICD sequences, while others are based solely on textual descriptions of the codes. This fundamental input difference makes direct comparisons between methods unfair. How can the authors justify such comparisons? Should a more balanced controlled experiment using the same input conditions be conducted?

---

### Official Review · Reviewer_k4Hi · 2025-10-27

**Soundness:** 2
**Presentation:** 2
**Contribution:** 2
**Rating:** 2
**Confidence:** 4

**Summary:**

The paper presents a systematic comparative study on uncovering interconnections between diseases using a wide range of statistical, pre-trained, and LLM-based methods. The authors analyze how different categories of models capture comorbidity and semantic relations among ICD-10 disease codes by constructing and comparing large interconnection matrices derived from both real-world EHR data (MIMIC-IV) and textual ICD descriptions. Experimental results show that domain-pretrained models such as Med-BERT align more closely with real co-occurrence patterns, while LLMs exhibit weaker and less clinically grounded associations.

**Strengths:**

(a) The paper conducts a comprehensive comparative study that evaluates how statistical, pretrained, and LLM-based methods uncover interconnections between diseases. By analyzing ten representative approaches across four methodological families, it provides unified evaluations bridging structured EHR data and text-based ICD information.

(b) The study combines multiple evaluation perspectives, including correlation analysis, graph-based comparison, and visualization of embedding spaces, to examine differences among models in capturing comorbidity patterns. This multi-angle analysis offers a clear view of how various modeling paradigms interpret disease relationships.

(c) Experiments are conducted on large-scale real-world clinical data (MIMIC-IV) and ICD-10 textual descriptions, with all interconnection matrices, visualizations, and the resulting consensus ontology released publicly. The work thus contributes a valuable benchmark resource for future studies on disease relationship modeling and LLM evaluation in healthcare.

**Weaknesses:**

(a) There are several typos and repeated sentences. For example, lines 102–106 in the Related Work section repeat the same sentence about Fotouhi et al. (2018), which indicates insufficient proofreading and weakens the overall presentation quality.
(b) The paper shows inconsistent descriptions and terminology. For instance, the abstract refers to seven approaches, while the main methodology section enumerates ten distinct methods (Fisher’s exact test, Jaccard similarity, MLM, Med-BERT, BioClinicalBERT, BERT, Yandex Doc Search, and three LLMs).

(c) The presentation of results lacks interpretive depth. Although many figures (heatmaps, t-SNE plots, graphs) are included, the discussion mainly restates observable patterns without deeper analysis of why the models behave differently or what kinds of relationships each model captures. Providing more qualitative case studies or concrete examples of clinically meaningful and spurious connections would make the conclusions more insightful and impactful.

(d) The paper lacks methodological clarity and standardization across the compared approaches. While ten methods are evaluated, the authors do not explain how their outputs are normalized or made comparable. Each method produces scores on different scales and distributions. For instance, the uniform 95th-percentile thresholding may bias the graph-based results, since denser methods (e.g., text-based models) inherently yield higher similarity values. Describing how such scale differences are handled or controlled would greatly improve the methodological transparency and reliability of the comparisons.

(e) The paper only performs a horizontal comparison (benchmark-style study) with limited standardization and analytical depth.

**Questions:**

(a) Could the authors clarify how the similarity or interconnection scores from different methods are normalized or made comparable across scales? For example, how are statistical scores (e.g., Fisher’s exact test) aligned with cosine similarities from embedding-based methods?

(b) When constructing the disease interconnection graphs using the 95th-percentile threshold, how sensitive are the results to this choice? Have the authors considered alternative thresholding strategies or normalization schemes to control for density differences between methods?

(c) The paper mentions that large language models (DeepSeek, Qwen, YandexGPT) are prompted using ICD-10 categories and descriptions. Could the authors provide examples of the prompts and discuss how prompt phrasing might affect the consistency or reliability of generated interconnections?

---

### Meta-Review · Area_Chair_ioSM · 2026-01-06

**Summary:**

The reviewers have several major concerns regarding the current paper.
For example, the lack of standardization among the compared methods raises concerns about the fairness of the evaluation outcomes and its reliability.
Issues that may arise based on the reliance on limited ICU dataset are not discussed properly.
The evaluation results reported in the manuscript lack interpretive depth and meaningful insights cross-checked by experts for validation.

**Reviewer Concerns:**

The authors have not responded to the reviewers questions and remarks, and all concerns remain unaddressed.

**Reviewer Scores:**

As the authors have not addressed any of the reviewers' concerns, the initial scores would remain unchanged.

---

### Decision · Program_Chairs · 2026-01-26

Reject